# A Deep Learning Approach to Detect Anomalies in an Electric Power Steering System

**DOI:** 10.3390/s22228981

**Published:** 2022-11-20

**Authors:** Lawal Wale Alabe, Kimleang Kea, Youngsun Han, Young Jae Min, Taekyung Kim

**Affiliations:** 1Department of AI Convergence, Pukyong National University, Nam-gu, Busan 48513, Republic of Korea; 2Department of Electric and Electronic Engineering, Halla University, Wonju 26404, Republic of Korea; 3Department of Computer & Information Technology, Incheon Jaeneung University, Dong-gu, Incheon 22573, Republic of Korea

**Keywords:** anomaly detection, deep learning, electric power steering, machine learning, sensor

## Abstract

As anomaly detection for electrical power steering (EPS) systems has been centralized using model- and knowledge-based approaches, EPS system have become complex and more sophisticated, thereby requiring enhanced reliability and safety. Since most current detection methods rely on prior knowledge, it is difficult to identify new or previously unknown anomalies. In this paper, we propose a deep learning approach that consists of a two-stage process using an autoencoder and long short-term memory (LSTM) to detect anomalies in EPS sensor data. First, we train our model on EPS data by employing an autoencoder to extract features and compress them into a latent representation. The compressed features are fed into the LSTM network to capture any correlated dependencies between features, which are then reconstructed as output. An anomaly score is used to detect anomalies based on the reconstruction loss of the output. The effectiveness of our proposed approach is demonstrated by collecting sample data from an experiment using an EPS test jig. The comparison results indicate that our proposed model performs better in detecting anomalies, with an accuracy of 0.99 and a higher area under the receiver operating characteristic curve than other methods providing a valuable tool for anomaly detection in EPS.

## 1. Introduction

Vehicle control systems have been significantly influenced by electrical power steering (EPS) due to the increased demand for environmental friendliness. In contrast to hydraulic power steering systems, in EPS systems, power is consumed only when assistance is required. This results in power efficiency since the EPS receives current from the alternator without direct connection with the vehicle engine (when the engine is powered).

EPS is typically preferred over hydraulic power steering for various reasons, including its ability to address leaking hydraulic hoses, its lower weight, and its potential for higher fuel savings [1,2]. With increasing use, EPS is becoming more sophisticated and widely adopted in the automobile industry. This necessitates enhanced safety, reliability, and performance [1,3] which can be achieved through implementing effective system design during the production process, enhancing anomaly detection, and implementing prognostics and health management of the EPS system.

An EPS system comprises various mechanical and electrical components, such as a torque sensor, handwheel angle sensor, speed sensor, and electronic control unit. These elements work together dynamically. Possible EPS system failures include anomalies or failures of components such as sensors and actuators, as well as impending failures such as insulation failure of the stator coil, damaged or defective bearings that impact friction [3]. Early detection of component failures is considered the most important type of failure detection, particularly of the incorrect steering caused by sensor malfunction. This type of failure can lead to catastrophic events due to driver panic [4]. However, the safety design for EPS has not been thoroughly investigated to avoid critical issues concerning the reliability, performance, and safety of EPS systems.

Anomaly detection, also known as “outlier” detection, can be defined as a technique to identify unexpected deviations of data points from typical patterns in a dataset. These deviations mainly occur due to gradual or sudden failures. The three main approaches to detecting this outlier are model-based, knowledge-based, and data-driven [5]. In recent years, the data-driven approach has demonstrated high potential with the development of deep learning models. The advent of Industry 4.0 has benefited industries by generating numerous datasets that can be utilized to monitor and improve production efficiency. This type of data, known as time-series data, has been widely adopted in forecasting and anomaly detection. However, obtaining data with an anomalous event is costly and difficult, which limits the use of the traditional data-driven method. The use of a deep learning algorithm for anomaly detection provides a solution to this challenge through an unsupervised or semi-supervised learning algorithm. The semi-supervised method is based on the assumption that all data used during training are normal data. In the testing stage, any data that deviates from the normal data are considered anomalies. In unsupervised learning, the state of the dataset to be used is undefined [6].

In this paper, we introduce a method for detecting anomalies of the EPS component by using deep learning with long short-term memory (LSTM) on raw sensor data. LSTM is a type of recurrent neural network (RNN) that mitigates the vanishing gradient problem by taking advantage of the temporal dependency of each time step [7,8]. The encoder compresses multidimensional data into a latent space while the decoder utilizes data in the latent space to reconstruct the representation of the input from the encoder. The reconstruction loss from training the normal dataset is used as a threshold for the anomaly score; that is, any value greater than the threshold during testing is classified as an anomaly. To evaluate the performance of our proposed method, we designed an EPS test jig and collected torque sensor data. Common sensor anomalies in EPS are caused by torque sensor and include difficulty turning the steering wheel, uneven left-right power steering assist, and a reduce amount of assistive torque when driving [9]. The main contributions of this paper as follows:Most existing methods for detecting anomalies in EPS components are physics-based modeling. In this paper, we propose a deep learning (data-driven) approach.We propose a two-stage approach for detecting anomalous scenarios in sample EPS data. Training is conducted using normal data and anomaly detection based on the reconstruction error.We utilized a dataset obtained experimentally from a test jig of an EPS system and compared the performance analysis to other methods used to detect anomalies.

The remainder of this paper is organized as follows:- Section 2 presents the related work used for detecting anomalies, while Section 3 describes our proposed method. Section 4 presents the experimental results, including the dataset used, model evaluation, and performance. Section 5 concludes the paper and describes future work.

## 2. Related Works

Recent advancements in sensors have increased the amount of data that can be captured and analyzed during the production process. These data can be used to improve decision-making for quality assurance and more effectively observe the manufacturing process [10]. Sensor data have enabled data-driven methods such as machine learning, which can be used for anomaly detection [11,12], intrusion detection [13], and quality prediction [14]. Data-driven defect detection is often associated with anomaly detection and is considered one of the most difficult tasks in machine learning. There are three methods of implementing data-driven defect detection: supervised, semi-supervised, and unsupervised learning methods. The method used depends on the characteristics of the available time-series data [5]. In this section, we introduce various methods for detecting anomalies, including the classical machine learning approach and deep learning.

### 2.1. Classical Machine Learning Anomaly Detection

Classical machine learning approaches for anomaly detection depend on the type of dataset and training algorithm; the datasets are either labeled, unlabeled, or partially labeled. Most supervised learning approaches for anomaly detection use label datasets as a binary classification task to distinguish between normal and anomalous data. Salman et al. [15] proposed linear regression and random forest to detect anomalies in a multi-cloud environment. They used a publicly available dataset to train and test the model for detection and categorization, and achieved a high accuracy of 99%. Deahyung et al. [16] proposed a data-driven classifier that identified representative anomalies from multimodal features. The classifier used conditional log-likelihood from sequences of input signals that are below a time-varying limit. However, this method often encountered the problem of class imbalance due to the smaller number of anomalies than the number of normal data in the training dataset [17]. Providing accurate labels, especially for the anomaly class, requires domain experts; as a result this process is time-consuming and expensive. Additionally, it can be challenging to scale high dimensional dataset. For this reason, unsupervised learning approaches are preferable.

In unsupervised learning, the spatial closeness of data points is utilized, which includes density-based and distance techniques to detect anomalies. Emadi et al. [18] proposed density-based spatial clustering applications with noise (DBSCAN) and support vector machine (SVM) to detect anomalies in wireless sensor networks using three features (voltage, humidity, and temperature). In their approach, it is necessary to evaluate the accuracy of the input parameters before using DBSCAN to detect anomalies. The authors assumed that two clusters are required for density-based detection of anomalies, and determined the relationship between input and output with coefficient correlation before labeling high-density clusters as normal. They then performed training with the SVM. Mistra et al. [19] presented a method for detecting outliers with local outlier factor (LOF) in a data stream. They assigned a real number to each data point, known as an outlier factor. The LOF value for data points that are deeply integrated into a cluster is given 1. For other points, the LOF value is larger than 1. Mistra et al. compared different LOF models and concluded that memory efficient incremental LOF algorithm is the most accurate and scalable in terms of detecting outliers in data streams.

Zhaolu et al. [20] proposed an integrated approach combining K-prototype clustering with K-nearest neighbor (KNN) classification algorithms to detect anomalies from massive system logs. This approach utilized a novel framework for feature extraction, clustering, and filtering. The authors examined the system based on session data and performed K-prototype clustering to divide the dataset into several groups depending on the extracted attributes to accurately define user activities. After filtering out normal occurrences, which typically appeared as highly coherent clusters, the remaining events are considered anomalies that warranted further investigation. The authors created two additional distance-based features to assess the local and global anomaly levels. They used KNN classifiers to evaluate the accuracy of their approach. In another study, Gerhard et al. [21] presented a novel traffic anomaly detection technique using K-means clustering. K-means clustering is a grouping approach that divides features into in K disjoint groups depending on their characteristic values [22]. In their study, unlabeled traffic flow records from training data are divided into clusters of regular and abnormal traffic. Then, matching cluster centroids are used as patterns for distance-based anomaly detection in the monitoring data.

Principal component analysis (PCA) is a method for reducing the dimensionality of datasets, improving interpretability, and minimizing information loss [23]. Takanori et al. [24] proposed a robust PCA method for detecting anomalies using daily or weekly periodicity in traffic volume. This method utilized a covariance matrix that is derived from normal traffic of the preceding period. The proposed method solved the problems of subspace contamination and a false negative ratio in anomaly detection. Amor et al. [25] introduced a method for detecting anomalies in medical measurements using PCA to examine physiological data gathered from sensors and identify multivariable abnormalities based on the squared prediction error in real-time. Their experimental results on an actual medical dataset revealed a high recall rate and low false positive rate.

### 2.2. Deep Learning-Based Anomaly Detection

Deep learning has been used to solve anomaly detection problems. Yunli et al. [26] introduced a deep learning approach for health monitoring and cooling equipment monitoring. Their method involved a two-stage approach that consisting of data prediction using an LSTM network and anomaly detection with the exponential weighted moving average using the prediction errors from the LSTM model. Zhao et al. [27] proposed an LSTM network for machine health monitoring. Their method provided promising results on raw sensor data with a shallow and deep LSTM network. Their experimental results indicated that the LSTM network is capable of learning meaningful representations from raw sensor signals in comparison to other deep learning model evaluated by the author.

Kim et al. [28] proposed convolution long short-term memory (C-LSTM) based web traffic anomaly detection to effectively model the spatial and temporal information in traffic data. They used an LSTM network to model temporal information, a convolution neural network to minimize the frequency fluctuations in spatial information, and a deep neural network for data mapping into a more separable space. They identified the best model by comparing their proposed approach with existing machine learning models through parametric trials, model comparison studies, and data analysis. There is a latency in detecting anomalies in real data because their proposed model used a sliding window to prepossess data. Anomaly detection in unsupervised multi-high dimensional datasets is frequently hindered by decoupled model learning with inconsistent optimization targets and the inability to preserve crucial information in low-dimensional space. Zong et al. [29] proposed a method of solving this problem using a deep autoencoding gaussian mixture model for unsupervised anomaly detection. The proposed model employed a deep autoencoder to generalize a low-dimensional representation and reconstruction error for each input data point, which is then fed into the gaussian mixture model.

Malhotra et al. [30] proposed the long short-term memory autoencoder model (LSTM-AE) to detect outliers using the reconstruct output of sensor data in mechanical devices. The training data contained only normal samples without outliers; and a higher reconstructed output in the test set sample data is an anomaly. Son et al. [31] reported the significance of structural health monitoring in ensuring the durability and safety of buildings and bridges. They presented a two-stage approach to anomaly detection of a cable-stayed bridge using an LSTM-autoencoder model. In the first stage, they classified the anomalies into continuous anomalies due to damaged structural damage and temporary anomalies due to inaccurate data. The cables causing temporary anomalies are selected, and the anomalous data are replaced by interpolated values. In the second stage, the authors examined the replaced inaccurate data to evaluate whether it is accurately categorized and substituted with acceptable values.

## 3. Workflow of EPS Anomaly Detection

This section introduces the overall architecture of our proposed method. Further details of the LSTM-autoencoder for anomaly detection are described in subsection three.

Figure 1 illustrates the framework of our proposed method for detecting anomalies in EPS data. We applied a deep learning model on time series torque data. The input data are preprocessed before being input into the LSTM-autoencoder model, and the model is trained with preprocessed data containing only normal samples from the collected datasets. The final stage of anomaly detection is based on reconstruction errors of the model on test samples that included both normal and anomalous samples.

### 3.1. Data Preprocessing

The general approach to deep learning is to investigate dataset for initial pattern discovery, handle missing or erroneous values, and normalize or standardize the data before feeding them to an anomaly detection algorithm. This approach is essential to prevent a data sample from skewing the objective function, especially in algorithms that compute the distance between features, if the datasets does not follow a normal distribution [32,33]. We scaled our dataset with normalization techniques in our proposed framework to prevent this scenario, using the scikit-learn MinMaxScaler to normalize values ranging from 0 to 1. The training, validation, and testing datasets are all scaled under the same conditions to validate that the actual value has not diverged from the scaling procedure. The equation for normalization is as follows:(1)$$Z_{i}=\frac{x_{i}-min_{x_{i}}}{max_{x_{i}}-min_{x_{i}}}$$
where $Z_{i}$ represents the normalized value, and $x_{i}$ indicate a data point from the actual dataset.

### 3.2. Model Training

#### 3.2.1. (LSTM)

The LSTM network is an improved form of RNN. The LSTM network provides a long-term memory cell to regulate the flow of information, capture long-term dependencies in time-series data, and determine the temporal correlations.. LSTM is an extension to RNN that allows for the use of long-term memory compared to ordinary RNN with short-term memory only [34]. There are various types of LSTM networks, with the most common one being the vanilla and peephole. The peephole networks have not been widely adopted in the literature, since recent studies provided contradictory results [35,36]. Working memory connections is introduced to achieve precise control of the gates and to address instability during training that is often encountered with peephole connections [36]. However, the proposed working memory connections model is limited due to the lack of significant performance improvement when training the stacked LSTM model. Our proposed framework adopts the popular vanilla network described in [35]. As illustrated in Figure 2, the vanilla LSTM is composed of a memory cell, input gate, output gate, and forget gate. The memory cell retains values over arbitrary time intervals, and the three gates control information flow in and out of the cell. Each of the gates is composed of a sigmoid layer followed by a pointwise multiplication operation. The sigmoid layer outputs integers ranging from zero to one, depending on which elements from the vector of the previous hidden state and new input data are allowed to enter the network.

Given input vector $x_{t}$ at time point *t* flowing into the network, the forget gate controls which information of the cell state is useful given the previously hidden state and new input data. The irrelevant component of the input is output as 0, while the relevant component is output as 1. The result of the forget gate can be mathematically expressed as follows:(2)$$f_{t}=\sigma \left(w_{f}\left[H_{t-1},X_{t}\right]+b_{f}\right)$$
where σ is the activation function, $w_{f}$ and $b_{f}$ are the weight and bias of the forget gate. The input gate has two primary objectives. First, it verify whether the new information (current input and previous hidden state) is important in the new cell state. Second, it updates the cell state. The input gate achieves these objectives in two stages. Similar to the forget gate, the first step is to determine which information is relevant in the current cell state. This process is defined as follows:(3)$$i_{t}=\sigma \left(w_{i}\left[H_{t-1},X_{t}\right]+b_{i}\right)$$
where σ is the activation function, and $w_{i}$ and $b_{i}$ are the weight and bias of the input gate. The next stage is to generates a memory cell $\tilde{C_{t}}$ by combining the previous hidden state with the current input data. In this process, a tanh activation function is used to generate the vector of the memory update, whose element values range from [1,−1]. This memory cell is defined as follows:(4)$$\tilde{C_{t}}=tanh\left(w_{c}\left[H_{t-1},X_{t}\right]+b_{c}\right)$$
where $w_{c}$ and $b_{c}$ are the weight matrices and the bias of the memory cell state. The two processes in Equations (2) and (3) are pointwise multiplied to update the old cell state $C_{t-1}$. Then, the combined operation of $C_{t-1}⊙f_{t}$ and addition by $i_{t}⊙\tilde{C_{t}}$ results in the new cell state $C_{t}$ of the network being updated, as defined in Equation (Equation 4):(5)$$C_{t}=f_{t}⊙C_{t-1}+i_{t}⊙\tilde{C_{t}}$$

The final gate is the output gate, which is updated with the new cell state. The output gate decides the new hidden state using the new cell state, previous hidden state, and current input data. The new input data and previous hidden state are fed through a sigmoid activated function to produce the output, as defined in Equation (Equation 5):(6)$$o_{t}=\sigma \left(w_{o}\left[H_{t-1},X_{t}\right]+b_{o}\right)$$
where $w_{o}$ and $b_{o}$ are the weight matrices and the bias of the output gate. As illustrated in Equation (Equation 6), the resulting output from filter vector $o_{t}$ and the cell state $c_{t}$, which is passed through the tanh activation function, are pointwise multiplied to obtain the new hidden state:(7)$$H_{t}=o_{t}⊙tanh\left(C_{t}\right)$$

The new hidden state $h_{t}$ and current cell states $c_{t}$ become the previous hidden state $h_{t-1}$ and previous cell state $c_{t-1}$ in the next LSTM unit. This process continues until the entire input sequence has been processed.

#### 3.2.2. Autoencoder

An autoencoder is a type of unsupervised neural network that efficiently encodes and decodes unlabeled input. The encoding process produces a latent representation of the inputs through dimensionality reduction, and then the decoder uses this representation to reproduce the original inputs [37]. As illustrated in Figure 3, the components of an autoencoder include an input layer, encoder, latent vector, decoder, and output layer. The objective of the encoder is to convert the input data *x* of a high dimensional vector $\left[x\in R^{m}\right]$ into a low-dimensional latent space representation *z* after removing insignificant data often known as noise, from the features. The input equation is as presented in Equation (Equation 7):(8)z=f(Wx+b)
where *f* is the activation function, and *W* and *b* are the weight and bias of low dimensional data sequence *z*, respectively. In the decoder, the latent vector is processed to generate the output $\hat{x}$ which is the reconstructed values of the input data.The output equation is as follows:(9)$$\hat{x}=f^{\prime }\left(W^{\prime }z+b^{\prime }\right)$$
where $f^{\prime }$ is the activation function, $W^{\prime }$ and $b^{\prime }$ are the weight and bias of the reconstructed input $\hat{x}$, respectively. In a typical autoencoder model, the reconstruction loss is minimized to reduce the difference between the input and output data. We used the mean absolute error (MAE) as our loss function to calculate the reconstruction loss. The reconstruction loss function is given is given as follows:(10)$$L\left(x-\hat{x}\right)=\frac{1}{n}\sum_{n=1}^{n}\left|\hat{x_{t}}-x_{t}\right|$$
where *n* is the number of samples in the training set, and *x* and $\hat{x}$ represent the input and output data, respectively. The autoencoder can be used as a feature extractor and fed into a supervised learning model, in addition to reducing the dimensionality of features [38]. Therefore, in our proposed model, we combined both the attributes of an autoencoder and LSTM. We employed the autoencoder to extract important features from data and inject them into the LSTM network to capture temporal dependencies.

### 3.3. Anomaly Detection

After training the model using normal datasets, the anomaly decision function can be used to evaluate the reconstructed test dataset and determine whether an anomalous event has occurred. As illustrated in Equation (Equation 10), we use MAE as the reconstruction loss. Other metrics for calculating the reconstruction loss include the mean square error; however, we use MAE since it has been proven in several studies to be more robust and minimize the false-positive rate in model predictions [39]. Our anomaly detection threshold is set as the maximum value of the MAE. Therefore, when sensor data are normal, reconstruction loss for the testing set is less than the anomaly threshold and more than the anomaly threshold when anomalies are present.

## 4. Experiment and Results

This section describes the experimental setup, including the data collection, training environment setup, and performance metrics. The results are also analyzed and discussed.

### 4.1. Data Collection

To experimentally verify the performance of our model, we collected torque sensor data from a test jig for an EPS using a 12-bit analog-to-digital converter. The experimental environment is illustrated in Figure 4. The running speed varied between 0 and 50 km/h, and the data acquisition board used is a field-programmable gate array. Through the board’s general purpose input and output pins, we transferred the sample data at 10-ms intervals. In total, we collected 80,000 samples.

### 4.2. Experimental Setup

We trained our model on a dataset of normal data, which has been demonstrated to provide satisfactory results with deep learning models in other studies [30,31,40]. Our test dataset included both anomalous and normal data. Before training the model, we divided the dataset into training, validation, and test sets. Training is performed using the training and validation sets to minimize the reconstruction loss. To evaluate the performance of our model training, we plotted the training and validation error versus the number of epochs, as illustrated in Figure 5. As can be seen, the loss stabilized quickly at approximately 10 epochs, and as expected, the validation loss stabilized thereafter. Table 1, presents the hyperparameters used for training the model. The encoder and decoder had two LSTM layers with 128 units each.

### 4.3. Evaluation Metrics

To evaluate the performance of the proposed methods, we used the classification accuracy (A), precision (P), recall (R), F-score (F), and area under the receiver operating characteristic curve (AUC). These are defined as follows:(11)$$A=\frac{\mathrm{TP}+\mathrm{TN}}{\mathrm{TP}+\mathrm{TN}+\mathrm{FN}+\mathrm{FP}}$$
(12)$$P=\frac{\mathrm{TP}}{\mathrm{TP}+\mathrm{FP}}$$
(13)$$\;R=\frac{\mathrm{TP}}{\mathrm{TP}+\mathrm{FN}}$$
(14)$$F1=2\;\cdot \;\frac{P\;\;\cdot \;R}{P+R}$$

True positive (TP) refers to the number of correctly detected anomalies, while true negative (TN) refers to the number of correctly detected normal data without anomalies. False positive (FP) refers to the number of normal data that are incorrectly classified as anomalies, while false negative (FN) refers to the number of anomalies that are not classified as anomalies. The accuracy of the anomaly detection algorithm is the proportion of correctly detected anomalies to all data in the dataset, while the precision is the ratio of true anomalies to predicted anomalies detected by the model.Recall is the percentage of anomalies predicted by the model out of the entire set of anomalies. The F-score is the harmonic mean of precision and recall. Therefore, the higher the accuracy is, the more accurate the model is at detecting normal and abnormal data. In addition, the higher the recall and precision are, the large the number of detected anomalies and the smaller the number of false alarms are, respectively. We conducted the experiment with three models on the EPS dataset and the metric scores are reported in Table 2.

### 4.4. Performance Results

#### 4.4.1. Anomaly Detection

The test sets are utilized to identify anomalies through inference from the trained model. The reconstruction error of the LSTM-AE model is used to measure the model’s performance. We set the anomaly threshold using Equation (Equation 10) and the method of scaling magnitude applied in [41] to simulate anomaly data of 501 samples from the test set. These samples are labeled as ground truth anomalies to evaluate the model’s anomaly detection capability. The maximum acceptable range of our EPS sensor data is illustrated in Figure 6, with the red dashed line representing the threshold and the red points representing the anomalies point. Generally, the data score are in the range of 0.00 and 0.01. Anomalies in sensors included drift, outlier, gap, and break anomalies, and the degree to which each anomaly type occurs varied. This is conducted by scaling the magnitude value from 1 to 1.5. A magnitude of 1 represents no anomalies and a magnitude of 1.5 indicates the value increased by 50%. We considered drift anomalies and outliers to be present when the overall distribution or individual points of the sample data changed, as this can be associated with mechanical structural failure or wear [42].

#### 4.4.2. Performance Analysis

To evaluate the performance of our model, we computed the confusion matrix and comparison table of the three models including our proposed model, gated recurrent unit autoencoder, and bi-directional longs short term memory autoencoder (LSTM-AE, GRU-AE and BiLSTM-AE). Figure 7 presents the confusion matrix for the test samples. The total number of test samples is 13,133. Of these, 12,632 are normal samples while 501 are abnormal samples. The LSTM-AE model identified 12,632 normal samples (accuracy: 0.99) and 492 abnormal data points(accuracy: 0.98) out of 501 abnormal samples. Our model had 0 false positives, indicating there was no inaccurate categorization of normal samples as abnormal. However, our model had nine false negatives, due to incorrectly classifying abnormal samples as normal. In the case of GRU-AE, 12,632 normal samples (accuracy: 0.99) are detected, while 446 abnormal samples (accuracy: 0.89) are detected out of the 501 samples. Furthermore, the Bi-LSTM model identified a total of 12,632 normal samples (accuracy: 0.99) and only 423 abnormal data points(accuracy: 0.84) out of the 501 abnormal samples.The performance of the models in summarized in Table 2.

Additionally, we plotted AUC to evaluate the classification accuracy of the three models in Figure 8. AUC indicates the degree of separability between the TP and FP rate of our model. The curve demonstrates the ability of our model (LSTM-AE) to detect anomalies by achieving a score of 0.99. The GRU-AE model score was 0.95, which was lower than that of the LSTM-AE, this is due to the absence of control memory cells. The output gate in the LSTM unit controls the amount of memory content that is accessed or used by other units in the network. The gated recurrent unit, on the other hand, discloses its entire content without any restriction. For the BiLSTM-AE, we obtained a score of 0.92, where the TP rate was observed to be low. The main reason for the difference in the performance of BiLSTM-AE and LSTM-AE is the direction of information. The BiLSTM-AE considers both directions when training the network, running the input from past to future and vice versa. It is computationally expensive to train a bidirectional LSTM, in addition, more data are required for the performance to be equivalent to that of a unidirectional LSTM. Table 3 assesses our model’s performance compared to similar techniques for anomaly detection that employ unique attributes of LSTM, AE, or a combination of both.

## 5. Conclusions

In this paper, we propose an LSTM-autoencoder-based anomaly detection framework for EPS sensor devices. The autoencoder uses an encoding function to capture the input representation and is implemented with an LSTM network to capture the temporal dependencies in the dataset. This method does not require expert knowledge of advanced feature engineering or complex preprocessing to discover meaningful features from the sensor signal. The proposed framework follows a semi-supervised learning approach that only requires normal data to train the model. Therefore, the proposed framework captures the expected distribution of the sensor signal. The maximum MAE of the reconstruction loss of the trained model is used as the anomaly score to detect anomalies during inference on the test data samples. We demonstrate that the proposed model can accurately detect the simulated drift and outlier anomalies.

This study only simulated two possible EPS sensor anomaly scenarios. In future work, we plan to investigate anomalous conditions collected from EPS systems, such as gaps anomaly when there is a high difference between primary and secondary signals of a non-contact torque sensor coil and abrupt drops from high to low of the speed sensor. To improve the robustness of the proposed framework for diagnosing EPS system anomalies.

## Figures and Tables

**Figure 1 sensors-22-08981-f001:**
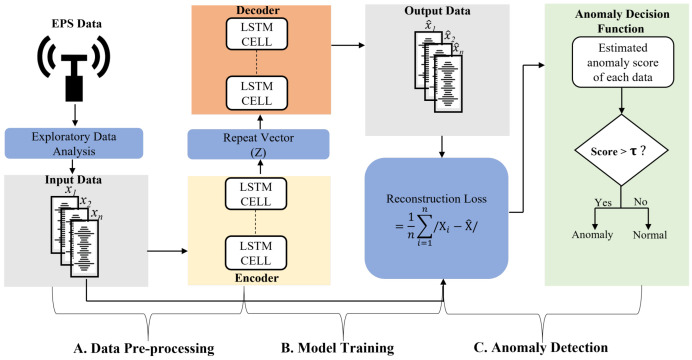
Framework of proposed anomaly detection model using electric power steering data.

**Figure 2 sensors-22-08981-f002:**
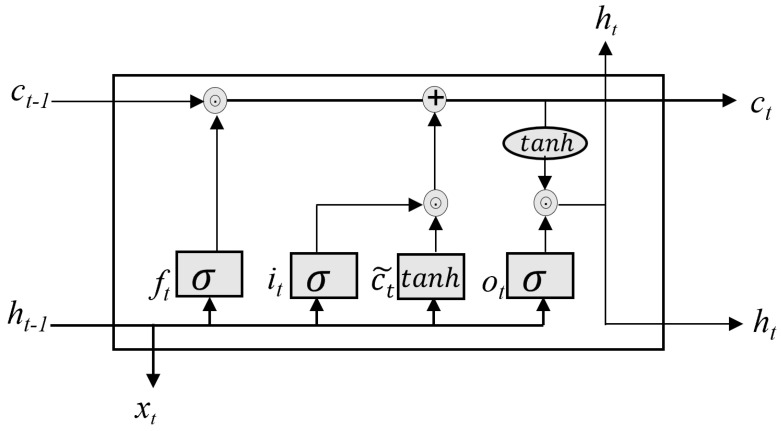
Schematic of long short-term memory (LSTM) model.

**Figure 3 sensors-22-08981-f003:**
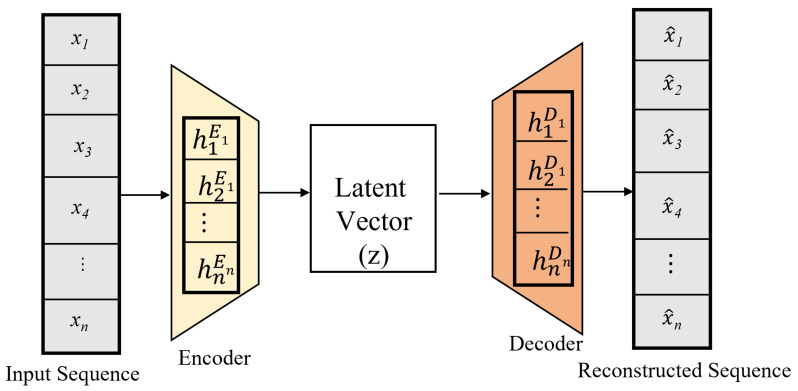
The Autoencoder Architecture.

**Figure 4 sensors-22-08981-f004:**
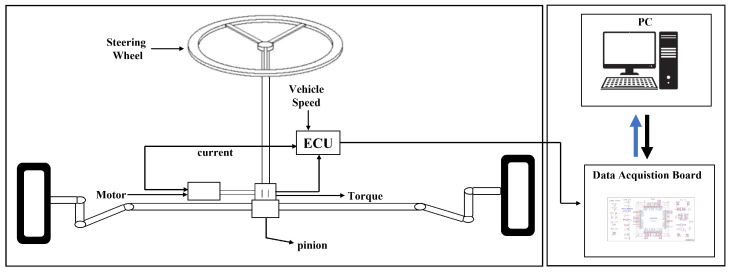
Data collection setup schematic.

**Figure 5 sensors-22-08981-f005:**
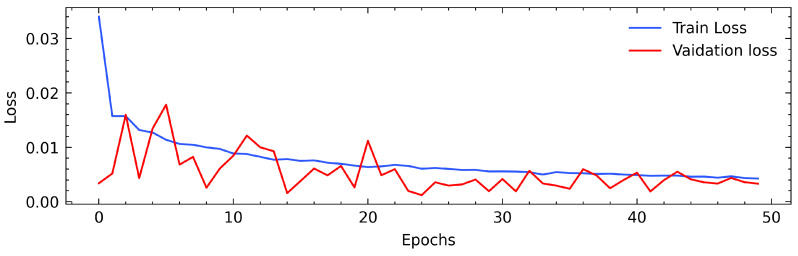
Training and validation loss over epochs.

**Figure 6 sensors-22-08981-f006:**
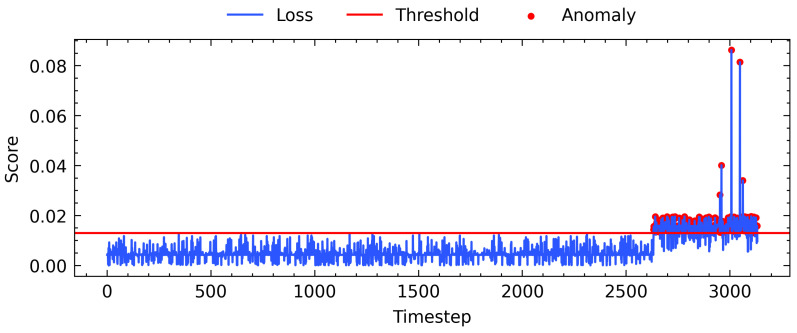
Training and validation Loss against number of epochs.

**Figure 7 sensors-22-08981-f007:**
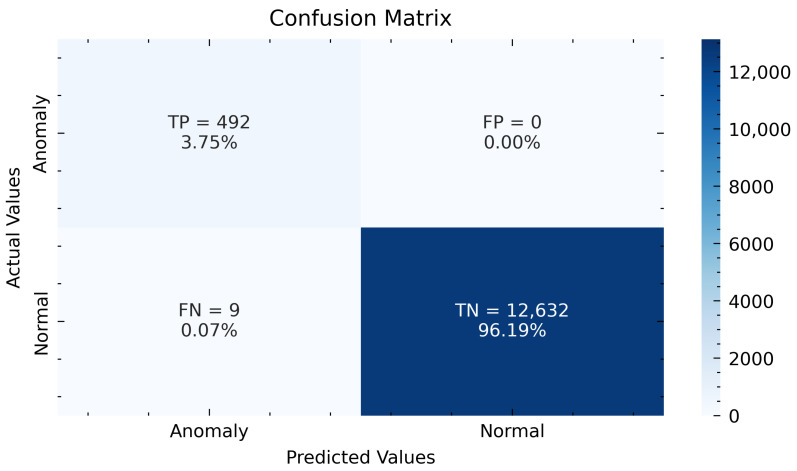
Model detection result from the confusion matrix.

**Figure 8 sensors-22-08981-f008:**
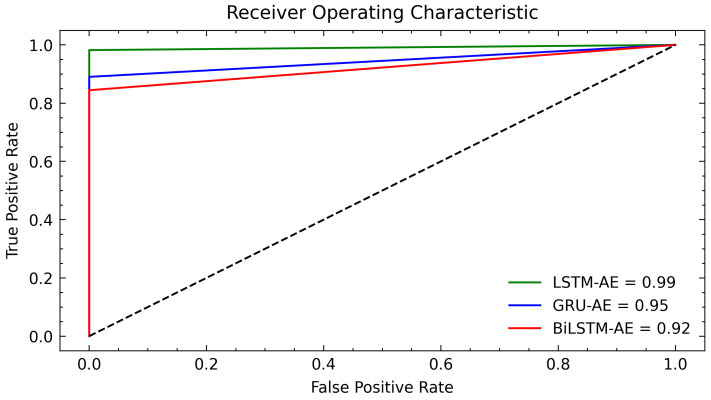
ROC curve for LSTM-AE, GRU-AE and BiLSTM-AE anomalies detection model.

**Table 1 sensors-22-08981-t001:** The hyperparameters used for model training.

Parameter	Value
Model Framework	PyTorch 1.12.1
Layers	2
Learning Rate	0.0009
Optimizer	Adam
Loss Function	MAE
Number of Epoch	50

**Table 2 sensors-22-08981-t002:** Performance comparison based on TP, FP, FN, TN value, detection accuracy, precision, recall and F1-score.

Model	TP	FP	FN	TN	Accuracy	Precision	Recall	F1-Score
BiLSTM-AE	423	0	78	12,632	0.9940	0.9999	0.8443	0.9155
GRU-AE	446	0	55	12,632	0.9958	0.9999	0.8902	0.9419
**LSTM-AE**	**492**	**0**	**9**	**12,632**	**0.9993**	**0.9999**	**0.9820**	**0.9809**

The bold text represents the model with the highest score in the performance analysis.

**Table 3 sensors-22-08981-t003:** Performance comparison with other comparable models.

Model	Source	Datasets	Accuracy	Precision	Recall	F1-Score
C-LSTM	[28]	Webscope S5	98.6	96.2	89.7	92.3
LSTM-AE	[31]	IPC-SHM2020	0.9998	0.9568	0.9201	0.9381
LSTM-AE	[38]	ECG	98.57	97.74	98.85	-
LSTM-AE	[39]	BOU	0.9444	0.9794	0.8577	0.9145
LSTM-AE	[43]	Solar plant generation	0.8963	0.9474	0.9432	0.9453
BILSTM-AE	[44]	Smart meter	0.9957	0.9958	0.9999	0.9978
BILSTM-VAE	[45]	UNM	90.01	84.59	97.87	90.75
LSTM-AE	Ours	EPS	0.9993	0.9999	0.9820	0.9809

## Data Availability

Not applicable.

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
