# Peer review of "A Deep Learning Approach to Detect Anomalies in an Electric Power Steering System"

_sensors, 2022, doi:10.3390/s22228981_

Round 1

Reviewer 1 Report

This study is described relatively specifically as a deep learning-based technology for EPS systems.

However, some supplements are needed.

1.  line 105 , 3.1 data preprocessing may be deleted.

2. For the understanding of subscribers,

In addition to the comparison results of the three types (LSTM-AE, GRU-AE, BiLSTM-AE) in Table 3, it is necessary to reflect the TP, TN, FP, and FN of each model.

3. A description of the right axis in Figure 7 should be displayed.

4. In conclusion, it is very important for the authors to mention a forward-looking and comprehensive plan for future experiments.

This is why the author's research is valuable.

Author Response

Dear Reviewer

We appreciate the timely, valuable, and constructive comments on our manuscript submitted to the MDPI Sensor Journal during the careful review process.

Reviewer Comments:

This study is described relatively specifically as a deep learning-based technology for EPS systems. However, some supplements are needed.

Author response: Thank you for the prompt feedback on the article. We appreciate your valuable input in our manuscript.

  1. line 105, 3.1 data preprocessing may be deleted.

Author Action:  The data preprocessing section has been modified in the revised version of our manuscript to improve its relevance in our proposed framework.

  1. For the understanding of subscribers,

In addition to the comparison results of the three types (LSTM-AE, GRU-AE, BiLSTM-AE) in Table 3, it is necessary to reflect the TP, TN, FP, and FN of each model.

Author Action:  Based on the reviewer's suggestions, Table 2 now reflects the values of TP, TN, FP, and FN of each model.

  1. A description of the right axis in Figure 7 should be displayed.

Author Action:  As the reviewer pointed out, the revised paper includes abbreviations and values in Figure 7.

  1. In conclusion, it is very important for the authors to mention a forward-looking and comprehensive plan for future experiments. This is why the author's research is valuable.

Author response:  The reviewer mentioned a vital keynote. Please accept our sincere thanks for this comment.

Author action: As the reviewer commented, we have added the summary of future work as detecting gaps anomaly when there is a high difference between primary and secondary signals of a non-contact torque sensor coil and abrupt drops from high to low of the speed sensor in EPS.

Best regards

Authors

Reviewer 2 Report

The article presents an interesting approach. Its well written

Author Response

Dear Reviewer

We appreciate the valuable and constructive comment on our manuscript submitted to the MDPI Sensor Journal during the careful review process.

Reviewer Comments:

The article presents an interesting approach. It's well written.

Author response:  Thank you for the prompt feedback on the article. We appreciate that you find it informative.

Best regards

Authors

Reviewer 3 Report

This manuscript is suggested to be accepted for pubilication without revision.

Author Response

Dear Reviewer

We appreciate the valuable and constructive comment on our manuscript submitted to the MDPI Sensor Journal during the careful review process.

Reviewer Comments:

This manuscript is suggested to be accepted for publication without revision.

Author response: Thank you for the prompt feedback on the article. We appreciate your suggestion.

Best regards

Authors

Reviewer 4 Report

Comments:

1         Abstract can be reduced to 200 words

2         The manuscript is written in the past tense (Avoid using was, were). This can be changed to present or present continuous tense.

3         Line 8: compressed feature – compressed features

4         Line 10: a proposed approach was – proposed approach is

5         Line 12: exhibited better performance that – exhibits better performance than

6         Line 35: give space after [4].

7         Line 94: label – labeled

8         Line 107: is utilized. – is utilized,

9         Line 110: what are three features?

10     Line 131: In another study,

11     Line 205 – 208: The information provided is contradictory

12     Line 213: The proposed model is limed to what? Explain.

13     Page 6, Line 7: Rewrite the sentence “The initialstage is determines which information is relevant in the current cell state, similar to theforget gate”.

14     Page 6, Line 13 and page 7, Line 3: italicize tanh

15     Line 237: rewrite the sentence “however, specifically use MAE because has been found to be more robust in many cases”.

16     Line 239- 241: Rewrite the sentence “Therefore, our assumption was that the reconstruction loss for the testing set was similar to the anomaly threshold when the sensor data were normal, and greater than the anomaly threshold when there was an anomaly”.

17     Line 245- 246: Rewrite the sentence “In addition, it analyzes and discusses the results”.

18     Line 263: In Table 2, presents - Table 2 presents

19     Line 282-287: Consider rewriting the sentences technically rather like a lab report “We used the test set to identify anomalies by inference of the trained model. We assessed the performance of the model by measuring the reconstruction error of the LSTM- AE model. We set the anomaly threshold using Equation (9) and used the method of scaling the magnitude applied in [42] to simulate the anomaly data of 501 samples from the test set. These samples were inserted into the test set and labeled as the ground truth anomalies to evaluate the model anomaly detection capability”.

20     Line 336: what are the other anomalies that can be considered for future work?

Author Response

Dear Reviewer

We appreciate the valuable and constructive comments on our manuscript submitted to the MDPI Sensor Journal during the careful review process.

Reviewer Comments Summarized:

  1. Some tense changes, punctuation corrections, and rewriting sentences.

Author response:  Thank you for the timely feedback on the article. We appreciate your valuable input in our manuscript.

Author action:  We have modified the manuscript tenses, and the mentioned sentences have been improved for better understanding and clarification.

  1. The abstract can be reduced to 200 words.

Author action:  As the reviewer suggested, we have modified the manuscript abstract to 200 words in the reversed version.

  1. What are other anomalies that can be considered for future work?

Author response: The review mentioned a vital keynote. We want to thank you for this comment.

Author action: As the reviewer commented, we have added the summary of future work as detecting gaps anomaly when there is a high difference between primary and secondary signals of a non-contact torque sensor coil and abrupt drops from high to low of the speed sensor in EPS.

Best regards

Authors

Round 2

Reviewer 4 Report

The paper may be published in sensors journal in its present form, the queries are answered by the authors  without any flaws